

# Co-located contemporaneous mapping of morphological, hydrological, chemical, and biological conditions in a 5th order mountain stream network, Oregon, USA

Adam S. Ward[1], Jay P. Zarnetske[2], Viktor Baranov[3,4], Phillip J. Blaen[5,6,7], Nicolai Brekenfeld[5], Rosalie Chu[8], Romain Derelle[9], Jennifer Drummond[5,10], Jan Fleckenstein[11,12], Vanessa Garayburu-Caruso[13], Emily Graham[13], David Hannah[5], Ciaran Harman[14], Jase Hixson[1], Julia L.A. Knapp[15,16], Stefan Krause[5], Marie J. Kurz[11,17], Jörg Lewandowski[18,19], Angang Li[20], Eugènia Martí[10], Melinda Miller[1], Alexander M. Milner[5], Kerry Neil[1], Luisa Orsini[9], Aaron I. Packman[20], Stephen Plont[2,21], Lupita Renteria[22], Kevin Roche[23], Todd Royer[1], Noah M. Schmadel[1,24], Catalina Segura[25], James Stegen[13], Jason Toyoda[8], Jacqueline Hager[22], Nathan I. Wisnoski[26], Steven M. Wondzell[27]

[1] O'Neill School of Public and Environmental Affairs, Indiana University, Bloomington, Indiana, USA

[2] Department of Earth and Environmental Sciences, Michigan State University, East Lansing, Michigan, USA

[3] LMU Munich Biocenter, Department of Biology II, Großhaderner Str. 2, 82152 Planegg-Martinsried, Germany

[4] Department of River Ecology and Conservation, Senckenberg Research Institute and Natural History Museum, 63571 Gelnhausen, Germany

[5] School of Geography, Earth & Environmental Sciences, University of Birmingham, Edgbaston. Birmingham. B15 2TT. UK

[6] Birmingham Institute of Forest Research (BIFoR), University of Birmingham, Edgbaston. Birmingham. B15 2TT. UK

[7] Yorkshire Water, Halifax Road, Bradford, BD6 2SZ

[8] Environmental Molecular Sciences Laboratory, Pacific Northwest National Laboratory, Richland, WA, USA

[9] Environmental Genomics Group, School of Biosciences, the University of Birmingham, Birmingham B15 2TT, UK

[10] Integrative Freshwater Ecology Group, Centre for Advanced Studies of Blanes (CEAB-CSIC), Blanes, Spain

[11] Dept. of Hydrogeology, Helmholtz Center for Environmental Research - UFZ, Permoserstraße 15, 04318 Leipzig, Germany

[12] Bayreuth Center of Ecology and Environmental Research, University of Bayreuth, 95440 Bayreuth, Germany

[13] Earth and Biological Sciences Division, Pacific Northwest National Laboratory, Richland, WA, USA

[14] Department of Environmental Health and Engineering, Johns Hopkins University, Baltimore, Maryland, USA

[15] Department of Environmental Systems Science, ETH Zürich, Zurich, Switzerland



[16] Center for Applied Geoscience, University of Tübingen, Tübingen, Germany

[17] The Academy of Natural Sciences of Drexel University, Philadelphia, Pennsylvania, USA

[18] Leibniz-Institute of Freshwater Ecology and Inland Fisheries, Department Ecohydrology, Müggelseedamm 310, 12587 Berlin, Germany

[19] Humboldt University Berlin, Geography Department, Rudower Chaussee 16, 12489 Berlin, Germany

[20] Department of Civil and Environmental Engineering, Northwestern University, Evanston, Illinois, USA

[21] Department of Biological Sciences, Virginia Polytechnic Institute and State University, Blacksburg, Virginia, USA

[22] Pacific Northwest National Laboratory, Richland, WA, USA

[23] Department of Civil & Environmental Engineering & Earth Sciences, University of Notre Dame, Notre Dame, IN

[24] Earth Surface Processes Division, U.S. Geological Survey, Reston, Virginia, USA

[25] Forest Engineering, Resources, and Management, Oregon State University Corvallis, OR, USA

[26] Department of Biology, Indiana University, Bloomington, Indiana, USA

[27] USDA Forest Service, Pacific Northwest Research Station, Corvallis, Oregon, USA.

*Correspondence to*: Adam S. Ward (adamward@indiana.edu)

**Abstract.** A comprehensive set of measurements and calculated metrics describing physical, chemical, and biological conditions in the river corridor is presented. These data were collected in a catchment-wide, synoptic campaign in Lookout Creek within the H.J. Andrews Experimental Forest (Cascade Mountains, Oregon, USA) in summer 2016 during low

discharge conditions. Extensive characterization of 62 sites including surface water, hyporheic water, and streambed sediment was conducted spanning 1st through 5th order reaches in the river network. The objective of the sample design and data acquisition was to generate a novel data set to support scaling of river corridor processes across varying flows and morphologic forms present in a river network.  The data are available at http://www.hydroshare.org/resource/f4484e0703f743c696c2e1f209abb842 (Ward, 2019)

**1 Introduction**

The exchange of water, solutes, particulate matter, energy, and biota between surface and subsurface domains (collectively "river corridor exchange") underpins a host of environmental functions and ecosystem services (e.g., Brunke and Gonser, 1997; Boulton et al., 1998; Harvey and Gooseff, 2015; Tonina and Buffington, 2009; Krause et al., 2011, 2017). These beneficial functions are primarily derived from the interactions between physical, chemical, and biological processes in the

river corridor (e.g., McDonnell et al., 2007; Boano et al., 2014; Ward, 2015; Bernhardt et al., 2017). In a recent review, Ward (2015) identified two key deficiencies that must be addressed to advance our predictive understanding of the



functioning of the river corridor. First, although the physical, chemical, and biological processes are co-evolved and known to be tightly coupled, they are seldom co-investigated. More comprehensive characterizations of physical-chemical-biological conditions are required to enable the study of coupled processes that span these sub-systems. Second, most comprehensive studies are conducted at single locations within an extensive river network and are limited in their range of
spatial and temporal scales.

Ward and Packman (Accepted) note these two main limitations of previous studies may result in a misattribution of cause and effect because only a subset of the relevant scales and variables are captured in the studies. As a result of these limitations, we currently have only a general understanding of river corridor exchange processes, thus limiting our ability to
predict these processes or the associated ecosystem functions across spatio-temporal scales relevant to water resource managers or policy makers who typical operate at river network scales (Krause et al., 2011). In response, we endeavoured to collect river corridor data that directly address the two limitations by acquiring simultaneous, multidisciplinary measurements distributed across a river network. The result is a novel river corridor data set documented herein. Specifically, this paper presents the collection of a synoptic-in-time, distributed-in-space characterization of physical,
chemical, and biological conditions in the river corridor of the 5th order Lookout Creek stream network within the H.J. Andrews Experimental Forest and Long-term Ecological Research site (Cascade Mountains, Oregon, USA). This data set presents new opportunities for exploring multi-scale, interacting river corridor patterns and processes.

## 2. Study location and campaign design

### 2.1 Study catchment

The H.J. Andrews Experimental Forest (HJA) is a 5th order catchment draining about 6,400 ha. The forest is located in the Western Cascades, Oregon, USA. Elevation in the basin ranges from about 410 to 1,630 m a.m.s.l., and the landscape is heavily forested, including old growth Douglas fir forests "(~400 yr old) and areas of younger regrowth forest after wildfire or was replanted after forest harvest. Additional detail about the climate, morphology, geology, and ecology of the site and region are well described by others (Dyrness, 1969; Swanson and James, 1975; Swanson and Jones, 2002; Jefferson et al.,
2004; Deligne et al., 2017).

Within the study catchment, there are three predominant landforms (Table 1; Figs. 1, 2). First, lower elevations are typically underlain by thermally weakened Upper Oligocene - Lower Miocene basaltic flows. These landforms are typified by highly dissected landscapes resulting from rapidly incising v-shaped valleys that are steep and narrow, with colluvium emplaced by
high energy hillslope failures and debris flows. Second, high elevations are typically underlain by plieocascade volcanics. These higher-elevations have well-defined, u-shaped valleys resulting from glacial processes, with cirques at the head of



valleys and highly compacted glacial tills filling the valley bottoms. Third, several deep seated earth flows are emplaced on the Upper Oligocene - Lower Miocene basaltic flows. These earth flow landforms typically lack well developed drainage networks, because they are too young to have developed large valleys and thus have minimal lateral constraint or visible bedrock along the streams.

The HJA has been the site of forest management, watershed and ecosystem research since it was established as a U.S. Forest Service research site in 1948, and has been one of the National Science Foundation's Long-term Ecological Research sites since 1980. As a result of these efforts and sustained commitment to data stewardship, the HJA hosts an extensive catalogue of data, maps, images, models, and software that are complementary to the data presented in this publication and provide

context within which these data can be interpreted (see HJA Data catalog at https://andrewsforest.oregonstate.edu/data ). For example, there are many complementary datasets of interest to readers of this manuscript, including stream discharge (HF004), stream chemistry (CF002), meteorological data (MS001), precipitation and dry deposition chemistry (CP002), aquatic invertebrate inventories (SA012, SA013, SA017), and soil properties and chemistry (SP001, SP006, SP026). We note these data are only a subset of the available information and encourage users of the data to explore the HJA data

catalogue for additional information.

## 2.2 Synoptic campaign design

This study was designed to replicate characterizations of the river corridor at a total of 62 sites spanning 1st through 5th order reaches in the HJA. Site selection was based on (1) the presence of flowing surface waters; (2) stratification across

stream orders; (3) coverage of the three major landform units in the HJA; and (4) accessibility of sites. All sampling of water and streambed sediment was conducted within the period 26-July through 3-Aug-2016 with no flow or precipitation events recorded during the sampling campaign, and all solute tracer experiments during the period 31-July through 12-Aug-2016 (again with no recorded flow or precipitation events).

In addition to broad spatial coverage of the river network, we selected 4 subcatchments for a more detailed characterization consisting of replication along the study reach at 4 to 6 locations per subcatchment. These 4 subcatchments were selected to have one subcatchment in the 3 predominant landforms in the study catchment, plus a fourth subcatchment located where a large debris flow scoured a section of the river corridor to bedrock in 1996 (Johnson, 2004). The objective of including 2 subcatchments in the low-elevation landform, was to provide a space-for-time comparison (i.e., WS01 and WS03 provide

two realizations of the same landform type at different states in response to the large debris flow that typifies a key geologic disturbance in the system).





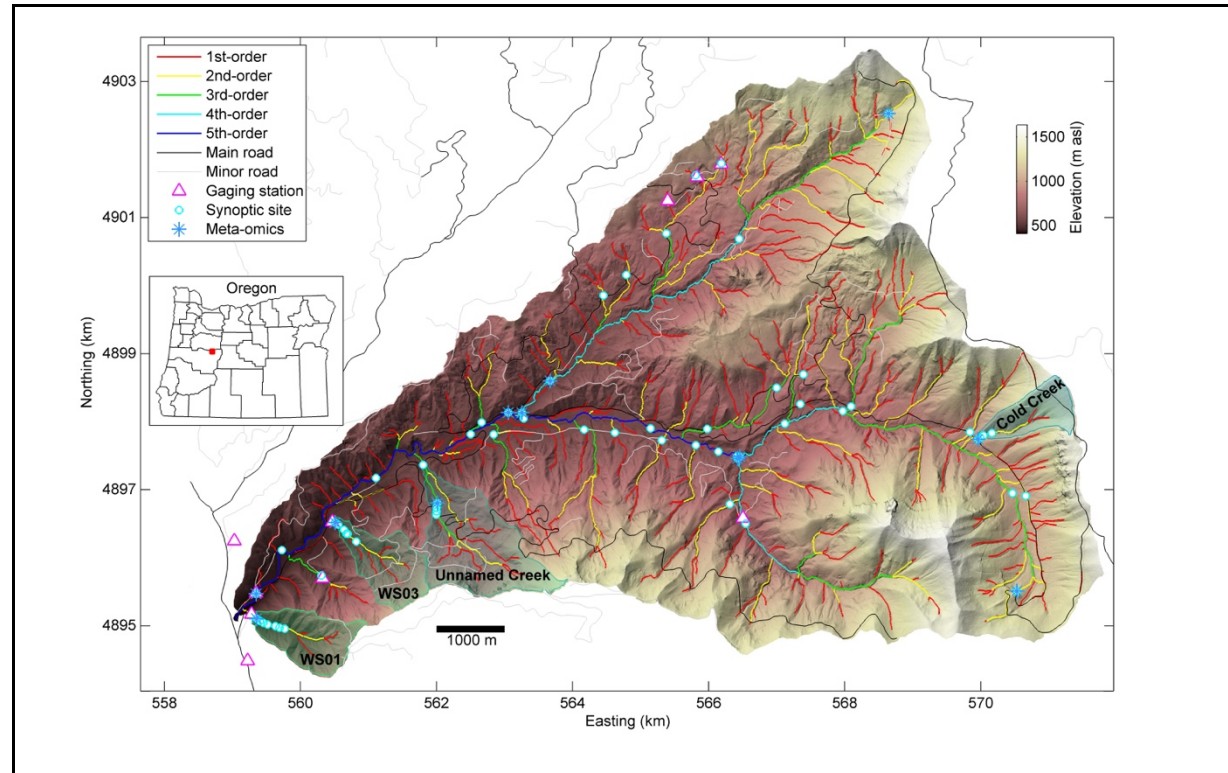

**Figure 1. Synoptic synoptic sites and LiDAR-derived stream network.**







**Figure 2. Headwater catchments in the major landform units at the H.J. Andrews Experimental Forest, including multiple synoptic sites along an intensively studied reach.**



**Table 1. Summary of site characteristics for the 4 headwater catchments where more intensive sampling was conducted. The descriptions of these headwater catchments are considered representative of the major landform types within the HJA [after Dyrness, 1969; Swanson and James, 1975; Swanson and Jones, 2002].**

| Site | Study Reach | Geologic Setting | Valley form | Colluvium presence and description | Notable river corridor description | Constraint | Lateral Inflows | Spatially Intermittant? |
|---|---|---|---|---|---|---|---|---|
| WS01 (HJA, Oregon) | Lower | Upper Oligocene - Lower Miocene Basaltic Flows, Volcanoclastic Rocks. Thermally altered (weakened) by subsequent volcanic activity enabling rapid downcutting of the valley bottoms. | V-shaped valley w/ Wide (10-20-m) valley bottom | Inceptisols. Abundant deposition from hillslope debris flows. Highly porous. Minimally compacted. | Pool-riffle-step and Pool-step-riffle morphology. Channel splits. Gravel wedges. Long, continuous sections of deposition from high-energy debris flow events. | Observed lateral (valley walls) and vertical (streambed) constraint of active channel | Proportional to lateral tributary area of hillslopes. Hillslopes underlain by intact bedrock. | Yes. Diurnal fluctuations in stream discharge enable rapid shift from continuous to intermittant over repeated 24-hr cycles. |
| | Middle | | | | | | | |
| | Upper | | | | | | | |
| WS03 (HJA, Oregon) | Lower | | V-shaped valley w/ Narrow (2-10 m) valley bottom | Deposition of colluvium from 1996 scouring event | | | | Yes |
| | Middle | | | Intermittant inceptisol-based colluvium on bedrock | Isolated gravel wedges formed by large woody debris | | | Yes, below features |
| | Upper | | | Minimal colluvium present | 100% Surface Flow (no colluvium) | | | No |
| Kerry Creek (HJA, Oregon) | Upper | Deep-seated earth failure on Upper Oligocene - Lower Miocene Basaltic Flows | Early downcutting & valley formation in unstructure colluvial material | Extensive colluvium. Flat and wide valley bottom with lateral meandering of active channel in incising valley bottom | Meanders, cut banks more typical of alluvial valleys compared to other study catchments proposed | No visible bedrock in active channel | No known groundwater nor lateral inflows. Minimal lateral tributary area in study reach | Unknown at this time. Expected due to the site of colluvial deposit. |
| | Lower | | | | | | | |
| Cold Creek (HJA, Oregon) | Upper | Plieocascase volcanics atop Middle and Upper Miocene Volcanics (Andesite, Basalt) | U-shaped valley (glacial cirque) | Compacted glacial tills | Large woody debris on till forms pools, steps with intermediate graveland cobble riffles | Bedrock visible at 1 location | Proportional to hillslope area | Unknown at this time. Not expected given apparent contributions from aqufier. |
| | Lower | | | | | | Aquifer extends beyond catchment | |



**Table 2. Left: Summary of sample collection (site characterization, streambed sediment, stream water, hyporheic water) and analyses included in this data set. Center: mapping of data types to their characterization of physical, chemical, and/or biological systems (after definitions of Ward, 2015). Right: data archival summary**



| | Sample Location | | | | System(s) Characterized | | | Data location |
| --- | --- | --- | --- | --- | --- | --- | --- | --- |
| | Site | Surface Water | Hyporheic Water | Streambed Sediment | Physical | Chemical | Biological | |
| Drainage area | ▓ | | | | ▓ | | | Tabular, Network Geometry |
| Valley slope | ▓ | | | | ▓ | | | Tabular, Network Geometry |
| Valley width | ▓ | | | | ▓ | | | Tabular, Network Geometry |
| Stream slope | ▓ | | | | ▓ | | | Tabular, Network Geometry |
| Stream width, depth | ▓ | | | | ▓ | | | Tabular |
| Stream order | ▓ | | | | ▓ | | | Tabular, Network Geometry |
| Sinuosity | ▓ | | | | ▓ | | | Tabular, Network Geometry |
| Discharge | ▓ | | | | ▓ | | | Tabular |
| Net gaining/losing | ▓ | | | | ▓ | | | Tabular |
| Site Coordinates | ▓ | | | | ▓ | | | Tabular |
| Temperature | | ▓ | ▓ | | ▓ | ▓ | | Tabular |
| Specific conductivity | | ▓ | ▓ | | ▓ | ▓ | | Tabular |
| 2H, 18O water isotopes | | ▓ | ▓ | | ▓ | ▓ | | Tabular |
| Hydraulic Conductivity | | | | ▓ | ▓ | | | Tabular |
| Grain size distirbution | | | | ▓ | ▓ | | | Tabular |
| DO | | ▓ | ▓ | | | ▓ | ▓ | Tabular |
| NPOC | | ▓ | ▓ | | | ▓ | | Tabular |
| SUVA254 | | ▓ | ▓ | | | ▓ | | Tabular |
| Spectral slope ratio | | ▓ | ▓ | | | ▓ | | Tabular |
| TDN | | ▓ | ▓ | | | ▓ | | Tabular |
| DOM EEMs | | ▓ | ▓ | | | ▓ | | Tabular |
| Fluorescence Index | | ▓ | ▓ | | | ▓ | | Tabular |
| Anions (Cl, SO4) | | ▓ | ▓ | | | ▓ | | Tabular |
| Cations (Na, K, Mg, Ca) | | ▓ | ▓ | | | ▓ | | Tabular |
| NO2+NO3 | | ▓ | ▓ | | | ▓ | | Tabular |
| PO4 | | ▓ | ▓ | | | ▓ | | Tabular |
| NH3 | | ▓ | ▓ | | | ▓ | | Tabular |
| Macroinvertebrate Community | | ▓ | | ▓ | | | ▓ | Tabular |
| EEA (N, P, C acquiring) | | | | ▓ | | ▓ | ▓ | Tabular |
| % Organic Matter | | | | ▓ | ▓ | ▓ | ▓ | Tabular |
| Stream solute tracer | ▓ | | | | ▓ | | | Solute Tracers |
| FT-ICR-MS | | | | ▓ | | ▓ | | FTICRMS |
| 16S Diversity DNA | | ▓ | ▓ | ▓ | | | ▓ | NCIB |



## 3. Methods

### 3.1 Synoptic Site Characterization

### 3.1.1 Topographic Analysis

The stream network was derived from a 1-m digital terrain model based on airborne LiDAR collected in 2008 (Spies, 2018).
We used the one-directional flow accumulation algorithm (Seibert and McGlynn, 2007) implemented in a modified version
of TopoToolbox (Schwanghart and Kuhn, 2010; Schwanghart and Scherler, 2014) to derive the direction of flow and
accumulation of drainage area within the basin. We defined the stream network as any location draining more than 5 ha. The
threshold was established based on iteratively comparing the derived stream network to our experience working in headwater
catchments and their extent (consistent with analyses by Ward et al., 2018). The TopoToolbox algorithm defined study
reaches as the segment between two junctions. In our analysis, we defined 686 river corridor segments including a total
length of about 209 km of valley containing about 242 km of stream. For each study reach, we tabulated the sinuosity of the
stream within the valley. Next, we discretized each reach into 10-m segments, extracting valley slope, stream sinuosity, and
stream slope for each segment (after Corson-Rikert et al., 2016; Ward et al., 2018). Each synoptic site was assigned a stream
order and average valley slope, streambed slope, and sinuosity for the reach within which it was located.

### 3.1.2 Hydraulic and valley geometry

At each synoptic site, field observations of valley width were collected using a tape measure, with valley edge being visually
defined in the field based on the hillslope break-point between the relatively flat valley bottom and steeper valley walls.
Total wetted channel width was measured perpendicular to the direction of flow at the synoptic site, and average channel
depth was recorded based on at least five measurements of depth spaced evenly across the channel.

### 3.1.3 Hydraulic conductivity

At the approximate centerline of the synoptic site, a Solinst 615N drive-point piezometer was driven to a depth of about 65-
cm below the streambed. The piezometer was screened over the distance of 50-65-cm below the streambed. The piezometer
was developed and purged by pumping slowly using a peristaltic pump until the water was visually clear, typically about 5
minutes. Then hyporheic water sampling occurred as described below (Section 3.2). Then a series of 3-6 replicates of a
falling head test were conducted using the piezometer, with water levels measured using a Van-Essen MicroDiver recording
at 0.5-s intervals and corrected for any variation in atmospheric pressure collecting data every 10-min. Falling head data
were used to estimate hydraulic conductivity after Hvorslev (1951). We report the geometric mean of the replicate tests for



each synoptic site. Finally, we note that at 5 sites there was minimal (∼<10cm) to no colluvium present in the valley bottom. At these sites we did not sample hyporheic water nor measure hydraulic conductivity, but we did collect streambed sediment from small in-channel deposits at the synoptic site. These sites are necessary for complete representation of the river corridor of the study catchment as there are many locations in the valley bottom that have minimal or no colluvium..

### 3.1.4 Macroinvertebrate community

Benthic macroinvertebrate colonization pots were installed at 44 of the 62 synoptic sites using the design of Crossman et al. (2012). Colonization pots were constructed of wire mesh (1.25 cm) cylinders approximately 15-cm in height and 8-cm in diameter. Hence, at sites where surface sediment grain sizes were larger than 8-cm, they could not be installed. Substrate was

excavated by hand and placed in each pot prior to installing so that the top of each pot was level with the streambed. Colonization pots remained in situ for about 6 weeks following installation. Removal was achieved by pulling a cable to raise a specially constructed tarpaulin bag around the sides of the pot before extraction, thereby minimizing sample loss. All substrate and macroinvertebrates were placed in a 90% ethanol solution for preservation. Additionally at 10 sites, surface samples of macroinvertebrates were collected with a Surber sampler with a 330 micron mesh net, collected in triplicate at

proximal locations and pooled for identification. Surface samples were processed using identical preservation methods; and identification was conducted by the same researcher.

After separation of macroinvertebrates, sediment samples were oven dried and sieved to assemble grain size distributions for each colonization pot. Importantly, because the pots were packed by hand in flowing water, we expect these grain size

distributions are biased toward the coarse fraction of streambed sediment, as finer materials would have washed away during packing, and also truncated as large cobbles would have been too large to pack into the pots and excluded from collection.

Identification was performed under the stereomicroscope, except for the Chironomidae larvae and early larval instars of the Plecoptera and Ephemeroptera, which were mounted in the Euparal and examined under the light microscope as described

by Andersen (2013). Macroinvertebrates were identified to the lowest possible taxonomic level, including the differentiation of adult and juvenile stages. Identification was performed using established keys (Merritt & Cummins, 1996; Andersen, 2013; Malicky, 1983; Langton, 1991; Epler, 2001).



### 3.2 Water sampling & analyses

### 3.2.1 Sample collection from stream and hyporheic zone

All water samples were collected using a peristaltic pump to sample water at a flow rate of about 0.5 L/min. The pump intake was located either in the stream thalweg for surface samples or in the developed piezometer for hyporheic samples.

Tubing was rinsed with water from the stream or hyporheic zone for at least 5 minutes prior to sample collection to minimize cross-contamination between sites.

First, water temperature and dissolved oxygen were recorded using a YSI ProODO handheld probe (YSI, Inc., Yellow Springs, OH, USA) with an optical dissolved oxygen (DO) sensor and thermistor. For stream samples, the probe was held in

the water column at the synoptic site near the pump intake.. For hyporheic samples, water was pumped into a small flow-through cell until it overflowed, and then the sensor placed into cell while flow continued. For both stream and hyporheic observations the sensor remained in place in the flowing water until probe readings for temperature and DO stabilized. Specific conductivity was also measured with a handheld conductivity probe (YSI EC300; YSI, Inc., Yellow Springs, OH, USA) using the same approaches.

Physical water samples for subsequent laboratory analyses were collected from the stream and hyporheic zone using identical methods, including: (1) Unfiltered samples for water isotope analysis (Section 3.2.2) were collected in 20 mL glass scintillation vials with conical inserts and were capped without headspace to minimize fractionation. (2) Samples for dissolved water chemistry and nutrients (Section 3.2.3) were collected by field filtering using handheld 65 mL syringes.

Syringes were triple rinsed with sample water prior to collection of any sample volume. Samples for dissolved organic carbon (DOC) analyses were field-filtered using a 0.2 μm cellulose acetate filter. Acid-washed amber HDPE bottles were triple-rinsed with filtered sample water prior to sample collection. DOC samples were placed in a cooler with ice in the field and remained chilled until analysis. Samples for dissolved nutrients, anions, and cations were field-filtered using a 0.45 μm cellulose acetate filter. Sample bottles were triple-rinsed with filtered sample water prior to sample collection. Dissolved

nutrient samples were placed on dry ice in the field immediately after collection and remained frozen until analysis. (3) Samples for microbial analysis (Section 3.2.4) were collected following Crevecoeur et al. (2015) by pumping water through a Sterivex (Millipore) cartridge with a 0.22 μm Durapore (PVDF) filter membrane until either 1 L of water was filtered or 45 minutes elapsed. Cartridges were immediately sparged to remove site water, filled with RNAlater stabilization solution (Ambion), and frozen in the field on dry ice. Samples remained frozen on dry ice until transferred and stored in a -80 °C

freezer until analysis.



### 3.2.2 Water stable isotopes ratios

We analyzed water stable isotopes to facilitate characterization of water ages using a cavity ring down spectroscopy method (Picarro L2130-I, Picarro Inc.), following laboratory protocols described by Nickolas et al. (2017). Briefly, samples were run under high-precision analysis mode using a 10 μL syringe for six injections per sample. We discarded the first three injections to eliminate memory effects. We used internal standards to develop calibration equations for stable isotopes of oxygen and hydrogen. The internal standards were calibrated using primary IAEA standards for Vienna Standard Mean Ocean Water (VSMOW2: $\delta18O = 0.0‰$, $\delta2H = 0.0‰$), Standard Light Antarctic Precipitation (SLAP2: $\delta18O = −55.5‰$, $\delta2H = −427.5‰$), and Greenland Ice Sheet Precipitation (GIPS: $\delta18O = −24.76‰$, $\delta2H = −189.5‰$). All stable isotopic values were reported as delta ($\delta$) values in parts per thousand (‰), which represent the deviation from the adopted VSMOW2 standard. Internal laboratory precision of the mean reported $\delta18O$ and $\delta2H$ values was estimated as 0.03‰ and 0.058‰ for $\delta18O$ and $\delta2H$ respectively based on the analysis of >50 duplicate samples. The external accuracy - representing the overall accuracy of the laboratory - was estimated as 0.058‰ and 0.241‰ for $\delta18O$ $\delta2H$ by comparing >60 estimated values for a known standard. A total of 7 samples collected for water isotope analysis were lost due to breakage of collection vials during transport. Paired surface- and hyporheic samples were re-collected on 1-3 August 2016 for these locations.

### 3.2.3 Dissolved water chemistry and nutrients

Dissolved nutrients $PO_4^{3-}$, $NO_2^-+NO_3^-$, and $NH_3$ were analyzed on a San++ Automated Wet Chemistry Analyzer - Segmented Flow Analyzer (Skalar Analytical B.V., Netherlands). Anions ($Cl^-$, $SO_4^{2-}$) and cations ($Na^+$, $K^+$, $Mg^{2+}$, $Ca^{2+}$) were analyzed on a Dionex ICS5000 ion chromatography system (Thermo Fisher Scientific). Samples were allowed to come to room temperature prior to analysis.

DOC concentrations (as non-purgeable organic carbon, NPOC) and total dissolved nitrogen (TDN) were analyzed via acid-catalyzed high temperature combustion using a Shimadzu TOC-L Analyzer with a TN module (Shimadzu Scientific Instruments, Kyoto, Japan). Samples were allowed to come to room temperature prior to analysis.

Dissolved organic matter (DOM) optical quality was analyzed via absorbance and fluorescence spectroscopy. UV-visible absorbance spectra ranging from 220 to 800 nm were collected using semi-micro, Brand-Tech cuvettes with a 1-cm path length on a Shimadzu dual-beam UV 1800 spectrophotometer (Shimadzu Scientific Instruments, Kyoto, Japan). Samples were allowed to come to room temperature prior to analyses. EPure water (18 MΩ, Barnstead EPure system) as a blank and cuvettes were triplicate rinsed with Epure water and rinsed with sample water between readings.



Excitation-Emission Matrices (EEMs) were measured over excitation wavelengths of 250-450 nm and emission wavelengths of 320-550 nm on a Horiba Aqualog Fluorometer (Horiba Scientific, Kyoto, Japan). Following the methods of Cory et al. (2010b), EEMs were generated for each sample using a 4 second integration time using a quartz cuvette with a 1-cm path length and Epure water as a blank. Samples were allowed to come to room temperature prior to analysis. Cuvettes were

rinsed with Epure water at least 10 times and triplicate rinsed with sample water between readings. EEMs were corrected for instrument-specific excitation and emission corrections and the inner-filter effect (Cory et al., 2010b). Epure water blank EEMs were collected and used to correct for Raman scattering. Fluorescence intensities from corrected-sample EEMs were converted to Raman units (Stedmon and Bro, 2008). EEMs corrections and processing were performed using Matlab consistent with Cory et al. (2010b).

Using EEMs and UV-visible absorbance spectra, several DOM quality indices were calculated for each sample. Specific UV absorbance at 254 nm (SUVA254) was calculated using absorbance readings at 254 nm normalized for path length (in m-1) and DOC concentration (in mg L-1). Higher SUVA254 values are associated with higher aromaticity of DOM (Weishaar et al., 2003). Spectral slope ratio (SR) was calculated from absorbance spectra following the methods of Helms et al. (2008).

SR values correspond inversely to relative DOM molecular weight. Fluorescence Index (FI) was calculated following Cory and McKnight (2005) as the ratio of emission (em) intensities for 470 nm and 520 nm at the 370 nm excitation (ex) wavelength. FI values correspond to DOM source with lower FI values corresponding to allochthonous, terrestrially-derived DOM and higher FI values corresponding to autochthonous, microbially-derived DOM (McKnight et al., 2001).

Intensities of specific EEMs peaks and absorbance wavelengths were selected and reported as well-documented proxies for character and sources of DOM. Following Coble (1996) and Cory and Kaplan (2012), EEMs peak A (ex 250, 420/em 500) and peak C (ex 250, 365/em 466) were reported as proxies for humic-like, terrestrially-derived fluorescent DOM (FDOM). EEMs peak T (ex 250, 285/em 344) was reported as a proxy for protein-like FDOM (Cory and Kaplan, 2012). Specific decadic and Naperian absorption coefficients reported serve as proxies for colored DOM (CDOM), and can be used as

indicators for specific sources and reactive fractions of the DOM pool (Spencer et al., 2009b). Decadic absorption coefficients (in m-1) were calculated from absorbance readings at specific wavelengths normalized for path length (in m). Naperian absorption coefficients (in m-1) are reported on a natural log scale and are calculated from absorbance readings at specific wavelengths normalized for path length (in m) and multiplied by a factor of 2.303.

**3.2.4 Microbial ecology**

To characterize the bacterial communities collected from the surface water and hyporheic zone, we first isolated the filter membrane from the Sterivex cartridge. We extracted DNA from the filters using the DNeasy PowerWater kit (Qiagen).



Following DNA extractions, we used PCR to amplify the V4-V5 region of the 16S rRNA gene using barcoded primers (515F and 806R) designed for the Illumina MiSeq sequencing platform (Caporaso et al. 2012). The sequence libraries were cleaned using the AMPure XP purification kit (Agencourt) and quantified using the PicoGreen dsDNA quantification kit (Quant-iT, Invitrogen). Libraries were pooled at 10 ng per library. Pooled DNA and Total RNA libraries were sequenced on the Illumina MiSeq platform at the Center for Genomics and Bioinformatics sequencing facility at Indiana University using paired-end reads (Illumina Reagent Kit v2, 500-reaction kit).

## 3.3 Sediment sampling & analyses

### 3.3.1 Sample collection

Streambed sediment samples were collected near the piezometer at each synoptic site. Sample collection involved manually removing the armor layer from the bed and then using a small specimen cup and putty knife to remove bed sediment without loss of fines. Samples were sieved to remove coarse material using a 2-mm sieve. Sieved material was placed in a sterile 50-mL centrifuge tube and frozen on dry ice immediately after collection. Samples were retained on dry ice or in a -80 °C freezer until analysis. Duplicate sediment samples were collected for analysis of extracellular enzymatic activity at 9 sites.

### 3.3.2 Extracellular Enzymatic Activity

Enzyme activities were determined using laboratory assays in which sediment extracts were exposed to model substrates that are hydrolyzed by the enzymes (Table 3). Protocols were based on those described by Sinsabaugh et al. (1997) and Belanger et al. (1997). Frozen sediment samples were thawed to room temperature and then 10 mL of 5-mM sodium bicarbonate buffer solution was added to approximately 1 mL subsamples of sediment in 15-mL centrifuge tubes. These tubes were homogenized with a vortex mixer for 15 s and then centrifuged for 15 min at 400 g. Samples were then stored in a refrigerator overnight and the following day 200 µL of the supernatant was pipetted in triplicate onto 96-well microplates. To ensure that any increase in fluorescence was due to enzyme activity, a set of control samples which had been boiled for 5 minutes to denature enzymes was also added to the plates. A set of standard solutions with known concentrations of fluorescent product were also added to each plate to generate a standard curve.

Background fluorescence readings were recorded and substrate solution was added to start the enzyme reaction. Each well in the microplate received 50 µL of a 200 µM substrate solution. Fluorescence measurements (440-nm emission intensity and 365-nm excitation wavelength) were recorded every ~30 min for at least 3 h. Microplates were protected from light and kept



at room temperature between readings. Fluorescence was measured using a BioTek Synergy Mx microplate reader. The accumulation of fluorescent products (AMC or MUF, see Table 2) from the hydrolysis reactions was measured over time and enzyme activity was calculated as the slope of a regression of AMC or MUF concentration against time.

5    About 1 mL of each sediment sample was dried, weighed, and then combusted at 550 oC and re-weighed to determine ash-free dry mass (AFDM) and percent organic content for the sample (Wallace et al. 2006). EEA rates were then normalized to organic matter content and are reported in units of μmol g AFDM-1 h-1.



**Table 3. Enzymes examined in this study and the reactions they catalyze.**

| Enzyme | Model Substrate | Product | Reaction |
|---|---|---|---|
| β-D-glucosidase (GLU) | 4-MUF-β-D-glucopyranoside | MUF[1] | Hydrolysis of glucose from cellobiose and cellulose |
| Alkaline phosphatase (AP) | 4-MUF-phosphate | MUF[1] | Hydrolysis of phosphate from phosphosaccarides and phospholipids |
| Leucine aminopeptidase (LAP) | L-Leucine -AMC | AMC[2] | Hydrolysis of leucine from polypeptides |
| N-acetylglucosaminidase (NAG) | MUF-N-acetyl-β -D-glucosaminide | MUF[1] | Degradation of chitin and other β-1,4-linked glucosamine polymers |

**1 MUF = 4-methylumbelliferyl**

**2 AMC = 7-amino-4-methylcoumarin**

5  **3.3.3 Organic matter characterization**

*FT-ICR-MS solvent extraction and data acquisition*

We use newly developed chemical extraction protocols combined with Electrospray ionization (ESI) and Fourier transform
ion cyclotron resonance (FT-ICR) mass spectrometry (MS) to infer differences in metabolites among our samples. ESI FT-
ICR-MS has emerged as a robust method for determining the chemistry of natural organic compounds (Kim et al., 2003;
10  Koch et al., 2005; Tremblay et al., 2007; Tfaily et al., 2011). ESI FT-ICR-MS has been used to distinguish metabolites
among ecosystems and soil types (Tfaily et al., 2015; Tfaily et al., 2017) as well as to provide information on the utilization
of distinct metabolites among samples within a single environment (Bailey et al., 2017; Graham et al., 2017; Stegen et al.,
2018).



Ultra-high resolution mass spectrometry of each sample was carried out using a 12 Tesla Bruker SolariX FT-ICR-MS located at the Environmental Molecular Sciences Laboratory (EMSL) in Richland, WA, USA. Prior to mass spectrometry, organic matter was extracted from sediments by adding 1 ml of water (18MΩ ionic purity) to 500 mg of sediments and shaking at 2000 rpm 20°C for 2 h on an Eppendorf Thermomixer (Tfaily et al. 2017). Samples were removed from the

shaker and left to stand before centrifugation at 2000 rpm for 10 min. The supernatant from each sample extraction was removed. Each sediment sample was extracted 3x with the above procedure. Supernatant from all extractions were combined and diluted to 5 mL to generate a final aliquot for analysis. These aliquots were acidified to pH 2 with 85% phosphoric acid and extracted with PPL cartridges (Bond Elut), following Dittmar et al. (2008). As per Tfaily et al. (2017), we performed weekly calibration using a tuning solution containing $C_2F_3O_2$, $C_6HF_9N_3O$, $C_{12}HF_{21}N_3O$, $C_{20}H_{18}F_{27}N_3O_8P_3$, and

$C_{26}H_{18}F_{39}N_3O_8P_3$ with mass-to-charge ratios (m/z) ranging from 112 to 1333 (Agilent Technologies, Santa Clara, CA USA), and instrument settings were optimized using Suwannee River Fulvic Acid (IHSS). The instrument was flushed between samples using a mixture of water and methanol. Blanks were analyzed at the beginning and the end of the day to monitor for background contaminants.

Samples were injected directly into the mass spectrometer and the ion accumulation time was was set to 0.1s . Data were collected from 98 – 900 m/z at 4M, yielding 144 scans that were co-added A standard Bruker electrospray ionization (ESI) source was used to generate negatively charged molecular ions. Samples were introduced to the ESI source equipped with a fused silica tube (30 μm i.d.) through an Agilent 1200 series pump (Agilent Technologies) at a flow rate of 3.0 μL min-1. Experimental conditions were as follows: needle voltage, +4.4 kV; Q1 set to 50 m/z; and the heated resistively coated glass

capillary operated at 180 °C.

*FT-ICR-MS data processing*

One hundred forty-four individual scans were averaged for each sample and internally calibrated using an organic matter homologous series separated by 14 Da (–CH2 groups). The mass measurement accuracy was less than 1 ppm for singly

charged ions across a broad m/z range (100-1200 m/z). The mass resolution was ~240K at 341 m/z. The transient was 0.8 seconds. Data Analysis software (BrukerDaltonik version 4.2) was used to convert raw spectra to a list of m/z values applying FTMS peak picker module with a signal-to-noise ratio (S/N) threshold set to 7 and absolute intensity threshold to the default value of 100. Peaks were treated as presence/absence data because peak intensity differences are reflective of ionization efficiency as well as relative abundance (Kujawinski and Behn, 2006; Minor et al., 2012; Tfaily et al., 2015;

Tfaily et al., 2017).

Putative chemical formulae were then assigned using in-house software following the Compound Identification Algorithm (CIA), proposed by Kujawinski and Behn (2006), modified by Minor et al. (2012), and previously described in Tfaily et al. (2017). Chemical formulae were assigned based on the following criteria: S/N >7, and mass measurement error <1 ppm,



taking into consideration the presence of C, H, O, N, S and P and excluding other elements. To ensure consistent formula assignment, we aligned all sample peak lists for the entire dataset to each other in order to facilitate consistent peak assignments and eliminate possible mass shifts that would impact formula assignment. We implemented the following rules to further ensure consistent formula assignment: (1) we consistently picked the formula with the lowest error and with the

lowest number of heteroatoms and (2) the assignment of one phosphorus atom requires the presence of at least four oxygen atoms.

The chemical character of thousands of peaks in each sample's ESI FT-ICR-MS spectrum was evaluated on van Krevelen diagrams. Compounds were plotted on the van Krevelen diagram on the basis of their molar H:C ratios (y-axis) and molar

O:C ratios (x-axis) (Kim et al., 2003). Van Krevelen diagrams provide a means to visualize and compare the average properties of organic compounds and assign compounds to the major biochemical classes (e.g., lipid-, protein-, lignin-, carbohydrate-, and condensed aromatic-like). In this study, biochemical compound classes are reported as relative abundance values based on counts of C, H, and O for the following H:C and O:C ranges; lipids ($0 < O:C \leq 0.3$, $1.5 \leq H:C \leq 2.5$), unsaturated hydrocarbons ($0 \leq O:C \leq 0.125$, $0.8 \leq H:C < 2.5$), proteins ($0.3 < O:C \leq 0.55$, $1.5 \leq H:C \leq 2.3$), amino sugars

($0.55 < O:C \leq 0.7$, $1.5 \leq H:C \leq 2.2$), lignin ($0.125 < O:C \leq 0.65$, $0.8 \leq H:C < 1.5$), tannins ($0.65 < O:C \leq 1.1$, $0.8 \leq H:C < 1.5$), and condensed hydrocarbons ($0 \leq 200$ $O:C \leq 0.95$, $0.2 \leq H:C < 0.8$) (Tfaily et al., 2015).

Finally, we calculated the Gibbs Free Energy of OC oxidation under standard conditions (ΔGoCox) from the Nominal Oxidation State of Carbon (NOSC) as per La Rowe and Van Cappellen (2011). NOSC was calculated from the number of

electrons transferred in OC oxidation half reactions and is defined by the equation:

$$NOSC = -((-Z + 4a + b - 3c - 2d + 5e - 2f)/a) + 4$$

where a, b, c, d, e, and f are, respectively, the numbers of C, H, N, O, P, S atoms in a given organic molecule and Z is net

charge of the organic molecule (assumed to be 1). In turn, ΔGoCox was estimated from NOSC following La Rowe and Van Cappellen (2011):

$$\Delta GoCox = 60.3 - 28.5(NOSC)$$

Values of ΔGoCox are generally positive, indicating that OC oxidation must be coupled to the reduction of a terminal electron acceptor. Though the exact calculation of ΔGoCox necessitates an accurate quantification of all species involved in every chemical reaction in a sample, the use of NOSC as a practical basis for determining ΔGoCox has been validated (Arndt et al., 2013; LaRowe and Van Cappellen, 2011; Graham et al., 2017; Boye et al., 2017; Stegen et al., 2018).



*Identification of putative biochemical transformations using FT-ICR-MS*

To identify potential biochemical transformations, we followed the procedure detailed by Breitling et al. (2006) and employed by Bailey et al. (2017), Graham et al. (2017), Graham et al. (2018), Moritz et al. (2017), Kaling et al. (2018), and Stegen et al. (2018). The mass difference between m/z peaks extracted from each spectrum with S/N>7 were compared to

commonly observed mass differences associated with biochemical transformations. All possible pairwise mass differences were calculated within each extraction type for each sample, and differences (within 1ppm) were matched to a list of 92 common biochemical transformations (e.g., gain or loss of amino groups or sugars). For example, a mass difference of 99.07 corresponds to a gain or loss of the amino acid valine, while a difference of 179.06 corresponds to the gain or loss of a glucose molecule. Pairs of peaks with a mass difference within 1 ppm of our transformation list were considered to be

related by the corresponding compound.

## 3.4 Stream solute tracer

Two injections of a conservative solute tracer (NaCl) were conducted at 46 synoptic sites, one each at the upstream and downstream reach boundaries to quantify discharge and short-term hyporheic flux. First, we fixed the upstream end of the

study reach at the same transect as the piezometer and sampling location. Next, we set the downstream station at a distance of about 20 wetted channel widths downstream from the piezometer and sampling location, a length selected to capture a representative valely segment (after Anderson et al., 2005). Minor variation in distance was allowed to place both sensors in well-mixed locations within the stream channel, with the total length reported for each tracer study reach. For each injection, mixing lengths for the solute tracer were visually estimated (after Payn et al., 2009; Ward et al., 2013b, 2013a), and small

releases of a visual tracer were used to confirm mixing lengths when visual estimates were uncertain. A known mass of NaCl was dissolved in stream water and released as an instantaneous injection one mixing length upstream from the reach boundary. Initially, the downstream slug was released and measured only at the downstream location to enable dilution gauging estimates of discharge at the downstream end of the study reach. Next, the upstream slug was released and monitored at both locations to enable dilution gauging at the upstream transect, and evaluation of both recovered and lost

tracer along the study reach. The experimental design closely follows Payn et al. (2009) and Ward et al. (2013b).

Solute tracer data at the reach boundaries were recorded as specific conductance (Onset Computer Corporation, Bourne, MA, USA). We used a four point calibration curve constructed by dissolving known masses of NaCl in stream water to convert specific conductance to salt concentration ($C = 0.5022S - 43.635$; where $C$ is NaCl concentration in mg/L and $S$ is

specific conductance; $r^2 > 0.99$). In addition to providing the full solute tracer timeseries, we also provide estimates of discharge (Q) based on dilution gauging, truncating the recovered tracer timeseries after 99% recovery (after Mason et al.,



2012; Ward et al., 2013b, 2013a). We report in the data set Q for both the upstream and downstream ends of the study reach, and the change in Q along the study reach.

## 4. Data Availability

5 These data are archived in the Consortium of Universities for the Advancement of Hydrologic Science, Inc. (CUAHSI) HydroShare data repository, accessible as http://www.hydroshare.org/resource/f4484e0703f743c696c2e1f209abb842 . In addition to tabular data, timeseries for solute tracer experiments and detailed results from the FT-ICR-MS analyses are archived. Raw sequence data for 16S DNA analyses are archived at the U.S. National Center for Biotechnology Information (NCBI) as a BioProject (Accession: #####).

## 5. Conclusions

We provide here a detailed characterization of physical, chemical, and biological parameters that are germane to the study of river corridor exchange and associated ecosystem functions and services. These data represent state-of-the-science characterization conducted at a heretofore unpresented resolution in space, and the only known data set that integrates across

15 physical, chemical, and biological dimensions of the river corridor, including coverage across 5 stream orders. Taken together, these data will enable the testing of hypothesized processes and relationships in the river corridor across spatial scales, and will be useful in the generation of testable hypotheses about river corridor exchanges in future studies.

**Author Contributions.**

20 All co-authors participated in the field collection, laboratory analysis, and/or curation of the data set. ASW was primarily responsible for the writing of this manuscript and assembly of the archival database. ASW and JPZ conceived of the study design with input from all co-authors. All authors contributed to the writing of this manuscript.

**Competing Interests.**

The authors report no conflicts of interest.



**Acknowledgements**

Funding for this research was provided by the Leverhulme Trust (Where rivers, groundwater and disciplines meet: a hyporheic research network), the UK Natural Environment Research Council (Large woody debris – A river restoration panacea for streambed nitrate attenuation? NERC NE/L003872/1), and the European Commission supported HiFreq: Smart

high-frequency environmental sensor networks for quantifying nonlinear hydrological process dynamics across spatial scales (project ID 734317), the US Department of Energy (DOE) Office of Biological and Environmental Research (BER) as part of Subsurface Biogeochemical Research Program's Scientific Focus Area (SFA) at the Pacific Northwest National Laboratory (PNNL). Data and facilities were provided by the HJ Andrews Experimental Forest and Long Term Ecological Research program, administered cooperatively by the USDA Forest Service Pacific Northwest Research Station, Oregon

State University, and the Willamette National Forest and funded, in part, by the National Science Foundation under Grant No. DEB-1440409. Ward's time in preparation of this manuscript was supported by the University of Birmingham's Institute of Advanced Studies. Additional support to individual authors is acknowledged from National Science Foundation (NSF) awards EAR 1652293, EAR 1417603, and EAR 1446328 and DOE award DE-SC0019377. Finally, the authors acknowledge this would not have been possible without support from their home institutions.

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
