# Peer review of "Co-located contemporaneous mapping of morphological, hydrological, chemical, and biological conditions in a 5th order mountain stream network, Oregon, USA"

_Earth System Science Data, 2019_

## Referee Comment (RC1) · Anonymous Referee #1 · 19 Jun 2019

Review of: Co-located contemporaneous mapping of morphological, hydrological, chemical, and biological conditions in a 5th order mountain stream network, Oregon, USA Ward et al. Summary: This field study was focused on an extensive 62 site, multi-day low flow sampling campaign across a 1st through 5th order river network. This study is unique in that it presents a physical, chemical, and biological dataset at a relatively high resolution spatial extent. This dataset will certainly be used extensively by this group and others to investigate spatial characteristics and drivers of riverine dynamics. My major comment is the lack of context for this study. The introduction is

short and does not present the state of the science for this type of research. As described in further detail in major comments below, I believe this manuscript will provide a larger impact in our community with the addition of a brief explanation of where our scientific community is in regards to our understanding of spatial physical, chemical, and biological characteristics of river networks. In addition to this, I included a handful of minor comments that I believe can improve the manuscript. Major Comments: Introduction: The introduction is short and leaves out important context. There have been a range of studies recently that have investigated spatiotemporal river network dynamics. These studies have mostly focused on hydrology or chemistry across river networks that range stream orders. This manuscript builds on those previous studies by incorporating not just hydrology and chemistry, but biology as well, in this spatial assessment. This manuscript and presentation of this dataset has the potential to be more impactful with a brief introduction of the current state of this work. See below for suggestions for several recent papers, although there are an extensive set of related papers on this topic: Hale, R. B., Godsey, S. (2019). Dynamic stream network intermittence explains emergent dissolved organic carbon chemostasis in headwaters. Hydrological Processes.

McGuire, K. J., Torgersen, C. E., Likens, G. E., Buso, D. C., Lowe, W. H., & Bailey, S. W. (2014). Network analysis reveals multiscale controls on streamwater chemistry. Proceedings of the National Academy of Sciences, 111(19), 7030–7035. https://doi.org/10.1073/pnas.1404820111 Zimmer, M. A., & McGlynn, B. L. (2018). Lateral, vertical, and longitudinal source area connectivity drive runoff and carbon export across watershed scales. Water Resources Research. Minor Comments: Abstract: Is there a reason the authors did not include information from results in the abstract? P 3 L 11: Replace "who typical" with "who typically" P 3 L 22: Remove " " " between "forests" and "(∼400". Section 2.1/Figure 1: How did the authors determine the first order streams? Is this based on the geomorphic channel network, or is this based on permanence of flow? Note: I later saw on P 10 Section 3.1.1. that stream orders were based on a topographic analysis with a 1m DEM (and potentially ground trothed).

[Figure]

Please make reference to this earlier. P 4 L 7: Should "term" be capitalized? Figure 1: It is difficult to differentiate the stream orders with the chosen elevation gradient. Perhaps the gradient can be grey scale to help the reader better identify the stream orders? Figure 2: Can the authors please label which of the four catchments represent which of the major landform units within Figure 2? Right now it is unclear which is which. P 10 L 24-29: It is unclear if the drive point piezometers were installed, purged, and hydraulic conductivity was measured all on the same day. If so, I am concerned that the piezometers were not collecting representative hydraulic conductivity values since the piezometers did not have time to "equilibrate" with the natural streambed. Further, if 3-6 replicates of the falling head test were done in sequential order, is it possible that the addition of water into the streambed may create zones of saturation, which may alter the hydraulic conductivity of the subsurface if it was previously dry. Did the authors see trends in the hydraulic conductivity measurements over the 3-6 replicates? If so, this may suggest these replicates were biased and a geometric mean is not the correct way to summarize the results. Honestly, I am surprised the authors could conduct a falling head test in a streambed – this suggests to me that the material below the streambed was dry, or there was perhaps a strong losing gradient. P 11 L 11-12: When where these pots installed? Were the installed during the synoptic sampling campaign, or taken out during the synoptic sampling campaign? This is important information, as 6 weeks is a large portion of the summer and macroinvertebrate communities may shift across these stream orders through the drying down of these river networks. P 12 L 5: Rinsing the tubing for 5 minutes with hyporheic water seems like it would greatly alter the hyporheic zone. 5 minutes of pumping at 0.5 L/min suggests that the authors extracted 2.5 L of hyporheic water before sample collection. That suggests the water that was sampled may be from preferential flowpaths that supplied water after the immediate region around piezometer was drained. P 13 L 12: Potentially missing "and" between "$\delta 18O$" and "$\delta 2H$". P 13 L 17-20: How quickly after sampling were these samples analyzed? How quickly did it take for samples to "come to room temperature"? Proper superscripts and subscripts needed for the dissolved

nutrients. P 15 L 6: What are EEA rates? I don't see this defined before in the text. P 18 L 15: Remove second "was" between "was" and "set" P 18 L 16: Potentially missing "." After "co-added" P 20 L 17: replace "valely" with "valley" P 20 L 17: The authors mention "to place both sensors", but what are the sensors used here? P 20 L 20: What is the approximate range of masses of NaCl used for this study? Are the metadata available for these dilution gauging experiments?

---

## Referee Comment (RC2) · Eric Moore (Referee) · 7 Aug 2019

Ward et al. Review

Co-located contemporaneous mapping of morphological hydrological, chemical, and biological conditions in a 5h order mountain stream network, Oregon, USA Ward et al.

Summary: The study involved intensive sampling of 62 field sites during baseflow conditions throughout a 5th order stream network. As noted in the introduction, this was a novel study looking at the interaction of physical, biological, and chemical variables

within the river corridor. The authors hosted the data publically to CUAHSI HydroShare which allows open access to the scientific community for use. The authors and other researchers are capable of using the data collected during this study for future analysis, publications, and repeatable studies. These data will be used to look at drivers of river corridor exchange and how they interact spatially throughout a network.

Comments: I'm assuming this is a data release manuscript, but I would like to see a bit more background in the introduction. I think this would help set up the "why" the authors collected the data when and how they collected it.

The Organic Matter Characterization method section gives too much detail and seems out of place with the other method sections. See my suggestions in the line-by-line review below.

Line by line review:

Page 1: Good

Page 2: Abstract - The abstract is short and concise but catches all the major topics of the paper Lines 26 - 28 - I suggest not having parenthesis in the first sentence of the article. Change the opening line to – River corridor science is the study of the exchange of water, solutes, particulate matter, energy, and biota between surface and subsurface domains, collectively called river corridor exchange.

Page 3: Lines 1 - 2 - Suggestion to switch around co-evolved and known to be tightly coupled. This allows the two co- words to be close together in the sentence and could help the reader understand the point more clearly. First, although the physical, chemical and biological processes are known to be tightly coupled and co-evolved, they are seldom co-investigated. Line 6 - Place the year into the Ward and Packman reference Lines 8 - 11 - Run-on sentence. Suggestion to change to - As a result of these limitations, we currently have only a general understanding of river corridor science exchange processes. This limits our ability to predict these processes or the associated ecosystem functions across spatio-temporal scales relevant to water resource managers and policymakers who typically operate at river network scales. Line 11 - change typical to typically Lines 14 - 17 - Good closing paragraph sentences. The last sentence in starting in line 16 could be bolstered up a little or moved before "Specifically,....". Using this as the last sentence in this paragraph defines the hard cut needed to go into the next section. Line 22 - 23 - Suggestion to change to - Elevation in the basin ranges from 410 to 1630 m, and the landscape is heavily forested with ∼400 year old Douglas fir trees with areas of younger forest from regrowth or replanting after timber harvest (∼400 yr old) is not really necessary if "old growth" is in front of it. I suggest using "including ∼400 year old Douglas fir forests"...." A.m.s.l (at mean sea level???) I don't think this is necessary

Page 4: Lines 20 - 24 – Split into two sentences All sampling of water and streambed sediment was conducted within the period 26-July through 3-Aug-2016 with no flow or precipitation events recorded during the sampling campaign. All solute tracer experiments occurred during the period 31-July through 12-Aug-2016, again with no recorded flow or precipitation events.

Page 5: Figure 1 – remove second synoptic from figure caption

Page 6: Figure 2 – Label each landform with a caption above each watershed's figure. There is no way for the reader to know which landform they are looking at.

Page 7: Table 1 – No need to repeat (HJA, Oregon) if all sampling happened there and it is listed in the table caption prior. I can see how this relates to Figure 2, but it would be great to see the creek names and landform types in Figure 2. This would help related the two figures better

Page 8: When printed out the caption of Table 2 appears on it's own page Reduce text size or table size to get Table 2's caption back together with Table 2

Page 9: Table 2: I really like this table! Suggestion to include units within this table

to show the wide range of data collected during this experiment. Having the units in the table would allow the reader to see what is comparable right away. See Page 8 comments to get the caption back together with Table 2

Page 10: None

Page 11: Lines 19 - 21 – Split this sentence into two sentences Lines 23 - 25 – Plecoptera and Ephemeroptera... Family? Genus???? Page 12: Line 8 – first reference from an instrument company. These references did not appear before this page

Page 13: None

Page 14: None

Page 15: Lines 13 -15: Description of sediment analysis method is well done, but what type of analyses were done on the sediment samples. Ash-free dry mass is listed a few paragraphs below, but what other sediment analyses were done?

Page 16: Configure Table 3 to fit beneath the paragraph on page 16.

Page 17: Configure Table 3 to fit beneath the paragraph on page 16. Line 6 – first subheading of the paper? Entire Organic matter characterization section needs to be shortened, cut, and less wordy. The background information from Lines 8 - 13 can all be covered with references.

Page 18: Shorten entire section Lines 9 - 10: write out correct chemical formulae or use chemical names with formulae in a table Lines 19 - 20: create a table of experimental conditions instead of listing them out. This is very out of place at the end of the paragraph. Line 25 – what is "the transient"? Definition of (m/z) is not clear when reading further down the page because there are too many acronyms within this section The suggestion of a table may help the reader keep things straight

Page 19: Shorten section Lines 13 - 16: Table of values would be more clear than writing them out

Page 20: Shorten section Lines 7 - 10 – Suggestion to remove a sentence that begins with "For example, . . .." This is not a method or needs to be described in a different way Line 12 – Valley spelled wrong Page 21: Good

Page 22: Good

Page 23: Good

Page 24: Good

Page 25: Good

Page 26: Good

Page 27: Good
* * *

---

## Author Response (AR1)

Referees' comments in bold type. Authors responses below each comment.

**Anonymous Referee #1**

**Review of: Co-located contemporaneous mapping of morphological, hydrological, chemical, and biological conditions in a 5th order mountain stream network, Oregon, USA Ward et al. Summary: This field study was focused on an extensive 62 site, multi-day low flow sampling campaign across a 1st through 5th order river network. This study is unique in that it presents a physical, chemical, and biological dataset at a relatively high resolution spatial extent. This dataset will certainly be used extensively by this group and others to investigate spatial characteristics and drivers of riverine dynamics. My major comment is the lack of context for this study. The introduction is short and does not present the state of the science for this type of research. As described in further detail in major comments below, I believe this manuscript will provide a larger impact in our community with the addition of a brief explanation of where our scientific community is in regards to our understanding of spatial physical, chemical, and biological characteristics of river networks. In addition to this, I included a handful of minor comments that I believe can improve the manuscript.**

> No response to the comments above, as this is a summary of more detailed points that are address individually below as "major comments".

**Major Comments:**
**Introduction: The introduction is short and leaves out important context. There have been a range of studies recently that have investigated spatiotemporal river network dynamics. These studies have mostly focused on hydrology or chemistry across river networks that range stream orders. This manuscript builds on those previous studies by incorporating not just hydrology and chemistry, but biology as well, in this spatial assessment. This manuscript and presentation of this dataset has the potential to be more impactful with a brief introduction of the current state of this work. See below for suggestions for several recent papers, although there are an extensive set of related papers on this topic:**
**Hale, R. B., Godsey, S. (2019). Dynamic stream network intermittence explains emergent dissolved organic carbon chemostasis in headwaters. Hydrological Processes.**
**McGuire, K. J., Torgersen, C. E., Likens, G. E., Buso, D. C., Lowe, W. H., & Bailey, S. W. (2014). Network analysis reveals multiscale controls on streamwater chemistry. Proceedings of the National Academy of Sciences, 111(19), 7030–7035. https://doi.org/10.1073/pnas.1404820111**
**Zimmer, M. A., & McGlynn, B. L. (2018). Lateral, vertical, and longitudinal source area connectivity drive runoff and carbon export across watershed scales. Water Resources Research.**

Accepted. We have modified the introduction to highlight the emerging class of spatially distributed observations in the river corridor space and the emergence of new techniques to interpret these data sets. Moreover, we clarify that our data set remains novel because it focused on simultaneous characterization of physical, chemical, and biological systems, spanning the stream water, hyporheic water, and sediment domains (in contrast to past studies that are primarily focused on in-stream water chemistry).

**Minor Comments:**
**Abstract: Is there a reason the authors did not include information from results in the abstract?**

Acknowledged. This primary purposes of this manuscript is to document a multi-scale, interdisciplinary data set that is available for the community. Thus, the primary results are the data themselves. We have elected to make no modifications in response to this comment.

**P 3 L 11: Replace "who typical" with "who typically"**

Accepted. Modified as suggested.

**P 3 L 22: Remove " " " between "forests" and "(_400".**

Accepted. Modified as suggested.

**Section 2.1/Figure 1: How did the authors determine the first order streams? Is this based on the geomorphic channel network, or is this based on permanence of flow? Note: I later saw on P 10 Section 3.1.1. that stream orders were based on a topographic analysis with a 1m DEM (and potentially ground trothed). Please make reference to this earlier.**

Accepted. We have added " (see details on network definition in section 3.1.1)." to the caption for Figure 1.

**P 4 L 7: Should "term" be capitalized?**

Accepted. Modified as suggested.

**Figure 1: It is difficult to differentiate the stream orders with the chosen elevation gradient. Perhaps the gradient can be grey scale to help the reader better identify the stream orders?**

Acknowledged. Greyscale is already used to represent roadways in the basin, which provide landmarks that link to other studies and maps in the basin. We have elected no modification in response to this comment.

**Figure 2: Can the authors please label which of the four catchments represent which of the major landform units within Figure 2? Right now it is unclear which is**

**which.**

> Accepted. We have added the following text to the Figure 2 caption: "WS01 and WS03 are located in the Upper Oligocene-Lower Miocene balsaltic flows, Unnamed Creek on a deep-seated earth flow, and Cold Creek in more modern Plieoscascade volcanics. Characteristics of each landform and catchment are detailed in Table 1."

**P 10 L 24-29: It is unclear if the drive point piezometers were installed, purged, and hydraulic conductivity was measured all on the same day. If so, I am concerned that the piezometers were not collecting representative hydraulic conductivity values since the piezometers did not have time to "equilibrate" with the natural streambed. Further, if 3-6 replicates of the falling head test were done in sequential order, is it possible that the addition of water into the streambed may create zones of saturation, which may alter the hydraulic conductivity of the subsurface if it was previously dry. Did the authors see trends in the hydraulic conductivity measurements over the 3-6 replicates? If so, this may suggest these replicates were biased and a geometric mean is not the correct way to summarize the results. Honestly, I am surprised the authors could conduct a falling head test in a streambed – this suggests to me that the material below the streambed was dry, or there was perhaps a strong losing gradient.**

> Acknowledged. Drive point piezometers were installed, purged, and measurements made on the same day. All falling head tests were conducted at locations with flowing surface stream water and were saturated prior to piezometer installation. All hydraulic conductivity replicates are provided in the tabular data for this study, and we do not observe a systematic shift in measurements as replicates proceeded (37 sites with positive trends, 20 sites with negative trends). We did not test the robustness of these trends with any statistical test given the comparable numbers in each direction and the small sample site at each site, but expect none would be statistically significant. Finally, falling head tests in streambeds are a common field technique. For example, see Baxter et al. and more than 200 articles citing this approach.

> Baxter, C. V.; Hauer, F.R.; Woessner, W.W. Measuring groundwater-stream water exchange: New techniques for installing minipiezometers and estimating hydraulic conductivity. *Trans. Am. Fish. Soc.* **2003**, *132*, 493–502.

> Ultimately, we made no edits in response to this comment.

**P11 L 11-12: When where these pots installed? Were the installed during the synoptic sampling campaign, or taken out during the synoptic sampling campaign? This is important information, as 6 weeks is a large portion of the summer and macroinvertebrate communities may shift across these stream orders through the drying down of these river networks.**

> Accepted. We have added "during the synoptic campaign" to the first sentence of section 3.1.4 to clarify the timing of installation. We also clarified the Surber samples were collected during the synoptic campaign.

**P 12 L 5: Rinsing the tubing for 5 minutes with hyporheic water seems like it would greatly alter the hyporheic zone. 5 minutes of pumping at 0.5 L/min suggests that the authors extracted 2.5 L of hyporheic water before sample collection. That suggests the water that was sampled may be from preferential flowpaths that supplied water after the immediate region around piezometer was drained.**

>   Accepted. We agree conceptually with this point. However, we did not control for nor record our flushing rate, though the lead author's recollection of the field campaign is that pumping was slower than the rate cited above. We do note that the sediment is highly porous and conductive, commonly coarse sands and gravels, where the flow disturbance may not be as significant as it would be in less hydraulically conductive material. Since we do not have data to present, we have acknowledged this in the study, adding "We did not record the pumping rates nor volumes for this rinse, and acknowledge it may have impact the flow field prior to sample collection. However, we expect this would be minimal because the sediment is generally highly hydraulically conductive."

**P 13 L 12: Potentially missing "and" between "_18O" and "_ 2H".**

>   Accepted. Modified as suggested.

**P 13 L 17-20: How quickly after sampling were these samples analyzed? How quickly did it take for samples to "come to room temperature"? Proper superscripts and subscripts needed for the dissolved nutrients.**

>   Accepted. We have modified the text to clarify this point as: Samples were thawed on the laboratory bench prior to analysis (typically 2-4 hours) and were analyzed at room temperature.

**P 15 L 6: What are EEA rates? I don't see this defined before in the text.**

>   Accepted. "EEA" replaced with "Extracellular enzymatic activity"

**P 18 L 15: Remove second "was" between "was" and "set"**

>   Accepted. Modified as suggested.

**P 18 L 16: Potentially missing "." After "co-added"**

>   Accepted. Modified as suggested.

**P 20 L 17: replace "valely" with "valley"**

>   Accepted. Modified as suggested.

**P 20 L 17: The authors mention "to place both sensors", but what are the sensors used here?**

Accepted. "Both sensors" replaced with "two specific conductivity sensors".

**P 20 L 20: What is the approximate range of masses of NaCl used for this study? Are the metadata available for these dilution gauging experiments?**

Acknowledged. NaCl masses are detailed in the tabular data set associated with this manuscript.

**Eric Moore (Referee #2)**
**eric.m.moore@uconn.edu**

**Ward et al. Review**
**Co-located contemporaneous mapping of morphological hydrological, chemical, and biological conditions in a 5h order mountain stream network, Oregon, USA Ward et al. Summary: The study involved intensive sampling of 62 field sites during baseflow conditions throughout a 5th order stream network. As noted in the introduction, this was a novel study looking at the interaction of physical, biological, and chemical variables within the river corridor. The authors hosted the data publically to CUAHSI HydroShare which allows open access to the scientific community for use. The authors and other researchers are capable of using the data collected during this study for future analysis, publications, and repeatable studies. These data will be used to look at drivers of river corridor exchange and how they interact spatially throughout a network.**

No response to the comments above, as this is a summary of more detailed points that are address individually below as "major comments".

**Comments: I'm assuming this is a data release manuscript, but I would like to see a bit more background in the introduction. I think this would help set up the "why" the authors collected the data when and how they collected it.**

See response to Referee #1 first major comment.

**The Organic Matter Characterization method section gives too much detail and seems out of place with the other method sections. See my suggestions in the line-by-line review below.**

See responses to suggested edits for pages 17-19, below.

**Line by line review:**
**Page 1: Good**

Off to a strong start just on the cover page and author list!

**Page 2: Abstract - The abstract is short and concise but catches all the major topics of the paper**

No response necessary.

**Lines 26 - 28 - I suggest not having parenthesis in the first sentence of the article. Change the opening line to – River corridor science is the study of the exchange of water, solutes, particulate matter, energy, and biota between surface and subsurface domains, collectively called river corridor exchange.**

Accepted. Modified as suggested.

**Page 3:**
**Lines 1 - 2 - Suggestion to switch around co-evolved and known to be tightly coupled. This allows the two co- words to be close together in the sentence and could help the reader understand the point more clearly. First, although the physical, chemical and biological processes are known to be tightly coupled and co-evolved, they are seldom co-investigated.**

Accepted. Modified as suggested.

Line 6 - Place the year into the Ward and Packman reference

Accepted. Modified as suggested.

**Lines 8 - 11 - Run-on sentence. Suggestion to change to - As a result of these limitations, we currently have only a general understanding of river corridor science exchange processes. This limits our ability to predict these processes or the associated ecosystem functions across spatio-temporal scales relevant to water resource managers and policymakers who typically operate at river network scales.**

Accepted. Modified as suggested.

**Line 11 - change typical to typically**

Accepted. Modified as suggested.

**Lines 14 - 17 - Good closing paragraph sentences. The last sentence in starting in line 16 could be bolstered up a little or moved before "Specifically,....". Using this as the last sentence in this paragraph defines the hard cut needed to go into the next section.**

Accepted. The sentence in question has been moved before "Specifically…" and now reads "The result is a novel river corridor data set documented herein that presents new opportunities for exploring multi-scale, interacting river corridor patterns and processes."

**Line 22 - 23 - Suggestion to change to - Elevation in the basin ranges from 410 to 1630 m, and the landscape is heavily forested with _400 year old Douglas fir trees with areas of younger forest from regrowth or replanting after timber harvest (_400 yr old) is not really necessary if "old growth" is in front of it. I suggest using "including _400 year old Douglas fir forests"...." A.m.s.l (at mean sea level???) I don't think this is necessary**

> Accepted. We have modified the "old growth" as suggested. We retain "a.m.s.l." (standard abbreviation for above mean sea level) for completeness.

**Page 4: Lines 20 - 24 – Split into two sentences All sampling of water and streambed sediment was conducted within the period 26-July through 3-Aug-2016 with no flow or precipitation events recorded during the sampling campaign. All solute tracer experiments occurred during the period 31-July through 12-Aug-2016, again with no recorded flow or precipitation events.**

> Accepted. Modified as suggested.

**Page 5: Figure 1 – remove second synoptic from figure caption**

> Accepted. Modified as suggested.

**Page 6: Figure 2 – Label each landform with a caption above each watershed's figure. There is no way for the reader to know which landform they are looking at.**

> Accepted. This information has now been added to the figure caption with an explicit cross-reference to Table 1, where details are provided.

**Page 7: Table 1 – No need to repeat (HJA, Oregon) if all sampling happened there and it is listed in the table caption prior. I can see how this relates to Figure 2, but it would be great to see the creek names and landform types in Figure 2. This would help related the two figures better**

> Accepted. We have removed the unnecessary "(HJA, Oregon), and the modifications to the Figure 2 caption now directly link the two elements, and we have added the following text to the table caption: "See catchment topography in Fig. 2 for each site."

**Page 8: When printed out the caption of Table 2 appears on it's own page Reduce text size or table size to get Table 2's caption back together with Table 2**

> Acknowledged. This will be addressed in typesetting of the final article. No modifications made at this time.

**Page 9: Table 2: I really like this table! Suggestion to include units within this table to show the wide range of data collected during this experiment. Having the units in the table would allow the reader to see what is comparable right away. See Page 8**

**comments to get the caption back together with Table 2**

> Acknowledged. Thanks for a great idea! We did attempt this, but some entries in the table are individual measures (e.g., stream width) while others actually describe a host of related and detailed observations (e.g., FT-ICR-MS). This made it difficult to avoid making a table that got too detailed and unwieldy, so we ultimately retained the form of the initial table.

**Page 10: None**

> ☺

**Page 11: Lines 19 - 21 – Split this sentence into two sentences**

> Accepted. Modified as suggested.

**Lines 23 - 25 – Plecoptera and Ephemeroptera: : : Family? Genus????**

> Accepted. "family" and "order" have been added to describe these and Chironomidae.

**Page 12: Line 8 – first reference from an instrument company. These references did not appear before this page.**

> Accepted. We have added documentation of which instrumentation were used to prior locations in the manuscript. Comparable information is now provided throughout the manuscript.

**Page 13: None**
**Page 14: None**

> Wahoo! Two in a row!

**Page 15: Lines 13 -15: Description of sediment analysis method is well done, but what type of analyses were done on the sediment samples. Ash-free dry mass is listed a few paragraphs below, but what other sediment analyses were done?**

> Accepted. We have added the following text to clarify the fate of these samples: "Samples collected in this fashion were used for extracellular enzymatic activity and FT-ICR-MS analyses, detailed in subsequent sections."

**Page 16: Configure Table 3 to fit beneath the paragraph on page 16.**
**Page 17: Configure Table 3 to fit beneath the paragraph on page 16.**

> Acknowledged. Page-break issues and layout will be finalized during the production process.

**Line 6 – first subheading of the paper? Entire Organic matter characterization section needs to be shortened, cut, and less wordy. The background information from Lines 8 - 13 can all be covered with references.**
**Page 18: Shorten entire section Lines 9 - 10: write out correct chemical formulae or use chemical names with formulae in a table**
**Lines 19 - 20: create a table of experimental conditions instead of listing them out. This is very out of place at the end of the paragraph.**
**Line 25 – what is "the transient"? Definition of (m/z) is not clear when reading further down the page because there are too many acronyms within this section**
**The suggestion of a table may help the reader keep things straight**
**Page 19: Shorten section Lines 13 - 16: Table of values would be more clear than writing them out**
**Page 20: Shorten section. Lines 7 - 10 – Suggestion to remove a sentence that begins with "For example, : : :." This is not a method or needs to be described in a different way**

Accepted. The section has been edited to remove extraneous details and streamline for the reader. Overall section 3.3.3 has been reduced by about 50% as a result of these edits.

**Line 12 – Valley spelled wrong**

Accepted. Modified as suggested.

**Page 21: Good**
**Page 22: Good**
**Page 23: Good**
**Page 24: Good**
**Page 25: Good**
**Page 26: Good**
**Page 27: Good**

Our best streak yet, albeit mainly the references!

**Co-located contemporaneous mapping of morphological, hydrological, chemical, and biological conditions in a 5th order mountain stream network, Oregon, USA**

Adam S. Ward[1], Jay P. Zarnetske[2], Viktor Baranov[3,4], Phillip J. Blaen[5,6,7], Nicolai Brekenfeld[5], Rosalie Chu[8], Romain Derelle[9], Jennifer Drummond[5,10], Jan Fleckenstein[11,12], Vanessa Garayburu-Caruso[13], Emily Graham[13], David Hannah[5], Ciaran Harman[14], Jase Hixson[1], Julia L.A. Knapp[15,16], Stefan Krause[5], Marie J. Kurz[11,17], Jörg Lewandowski[18,19], Angang Li[20], Eugènia Martí[10], Melinda Miller[1], Alexander M. Milner[5], Kerry Neil[1], Luisa Orsini[9], Aaron I. Packman[20], Stephen Plont[2,21], Lupita Renteria[22], Kevin Roche[23], Todd Royer[1], Noah M. Schmadel[1,24], Catalina Segura[25], James Stegen[13], Jason Toyoda[8], Jacqueline Wells[22], Nathan I. Wisnoski[26], Steven M. Wondzell[27]

[1] O'Neill School of Public and Environmental Affairs, Indiana University, Bloomington, Indiana, USA

[2] Department of Earth and Environmental Sciences, Michigan State University, East Lansing, Michigan, USA

[3] LMU Munich Biocenter, Department of Biology II, Großhaderner Str. 2, 82152 Planegg-Martinsried, Germany

[4] Department of River Ecology and Conservation, Senckenberg Research Institute and Natural History Museum, 63571 Gelnhausen, Germany

[5] School of Geography, Earth & Environmental Sciences, University of Birmingham, Edgbaston. Birmingham. B15 2TT. UK

[6] Birmingham Institute of Forest Research (BIFoR), University of Birmingham, Edgbaston. Birmingham. B15 2TT. UK

[7] Yorkshire Water, Halifax Road, Bradford, BD6 2SZ

[8] Environmental Molecular Sciences Laboratory, Pacific Northwest National Laboratory, Richland, WA, USA

[9] Environmental Genomics Group, School of Biosciences, the University of Birmingham, Birmingham B15 2TT, UK

[10] Integrative Freshwater Ecology Group, Centre for Advanced Studies of Blanes (CEAB-CSIC), Blanes, Spain

[11] Dept. of Hydrogeology, Helmholtz Center for Environmental Research - UFZ, Permoserstraße 15, 04318 Leipzig, Germany

[12] Bayreuth Center of Ecology and Environmental Research, University of Bayreuth, 95440 Bayreuth, Germany

[13] Earth and Biological Sciences Division, Pacific Northwest National Laboratory, Richland, WA, USA

[14] Department of Environmental Health and Engineering, Johns Hopkins University, Baltimore, Maryland, USA

*This draft manuscript is distributed solely for purposes of scientific peer review. Its content is deliberative and predecisional, so it must not be disclosed or released by reviewers. Because the manuscript has not yet been approved for publication by the U.S. Geological Survey (USGS), it does not represent any official USGS finding or policy.*

[revised manuscript text omitted]